# Defining the minimal components of the influenza A virus replication machinery via an in vitro reconstitution system

**Zihan Zhu, Haitian Fan¤, Ervin Fodor** *

Sir William Dunn School of Pathology, University of Oxford, Oxford, United Kingdom

¤ Current address: Department of Biophysics, School of Basic Medical Sciences, Zhejiang University School of Medicine, Zhejiang University, Hangzhou, China

* ervin.fodor@path.ox.ac.uk

**Data Availability Statement:** All relevant data are within the paper and its Supporting Information files.

**Funding:** This work was supported by Medical Research Council (MRC) programme grant MR/

## Abstract

During influenza A virus infection, the viral RNA polymerase transcribes the viral negative-sense segmented RNA genome and replicates it in a two-step process via complementary RNA within viral ribonucleoprotein (vRNP) complexes. While numerous viral and host factors involved in vRNP functions have been identified, dissecting the roles of individual factors remains challenging due to the complex cellular environment in which vRNP activity has been studied. To overcome this challenge, we reconstituted viral transcription and a full cycle of replication in a test tube using vRNPs isolated from virions and recombinant factors essential for these processes. This novel system uncovers the minimal components required for influenza virus replication and also reveals new roles of regulatory factors in viral replication. Moreover, it sheds light on the molecular interplay underlying the temporal regulation of viral transcription and replication. Our highly robust in vitro system enables systematic functional analysis of factors modulating influenza virus vRNP activity and paves the way for imaging key steps of viral transcription and replication.

## Introduction

Influenza viruses contain a segmented negative-sense RNA genome with each viral RNA (vRNA) segment assembled into an individual viral ribonucleoprotein (vRNP) complex [1]. In the vRNP, the vRNA segment forms a pseudocircle with the termini bound by a single copy of the vRNA-dependent RNA polymerase and the rest of the RNA encapsidated by an oligomer of viral nucleoprotein (NP). During viral infection, vRNPs are delivered into the host cell nucleus where they first act as templates for transcription producing capped and polyadenylated viral mRNA. Once viral proteins are translated and viral polymerase and NP accumulate in the nucleus, vRNPs switch to act as templates for genome replication. Both transcription and replication of vRNA are performed by the viral polymerase and require different modes of initiation and termination [2,3]. Transcription is a primer-dependent process, requiring cap-snatching activity of the polymerase, which is dependent on the association of vRNPs with host RNA polymerase II [4]. This association enables the viral polymerase to access nascent

R009945/1 (to E.F.) and Clarendon Fund and
Wellcome Infection, Immunity and Translational
Medicine studentship (to Z.Z.). The funders had no
role in study design, data collection and analysis,
decision to publish, or preparation of the
manuscript.

**Competing interests:** The authors have declared
that no competing interests exist.

**Abbreviations:** ANP32, acidic nuclear
phosphoprotein 32; cRNA, complementary RNA;
EMSA, electrophoretic mobility shift assay; FBS,
foetal bovine serum; LCAR, low complexity acidic
region; LRR, leucine-rich repeat; MDBK, Madin-
Darby bovine kidney; MOI, multiplicity of infection;
NA, neuraminidase; NP, nucleoprotein; NS,
nonstructural; PB1, polymerase basic 1; PB2,
polymerase basic 2; PMSF, phenylmethylsulfonyl
fluoride; rNTP, ribonucleotide triphosphate; SDS-
PAGE, sodium dodecyl sulfate polyacrylamide gel
electrophoresis; vRNA, viral RNA; vRNP, viral
ribonucleoprotein; WT, wild type.

host capped RNAs that are cleaved by the viral polymerase to generate a capped RNA fragment that acts as primer for transcription. Transcription terminates prematurely by the polymerase stuttering on a sequence of U residues close to the end of the vRNA template. In contrast, replication is a two-step primer-independent process, requiring de novo initiation by the polymerase, and proceeds to the end of the template, producing a complete copy of the template. In the first step, complementary RNA (cRNA) is produced, which acts as template for vRNA synthesis in the second step. vRNA and cRNA are assembled with polymerase and NP into vRNPs and cRNPs, respectively, in a process that is believed to take place in a co-replicational manner [5]. Members of the acidic nuclear phosphoprotein 32 (ANP32) family of host proteins are essential factors for influenza virus genome replication [6,7]. They have been proposed to support influenza virus genome replication in 2 ways. First, the N-terminal leucine-rich repeat (LRR) domain of ANP32 bridges a dimer formed by the resident polymerase from the vRNP and a newly translated viral polymerase, providing a platform for viral genome replication [8]. Second, the C-terminal low complexity acidic region (LCAR) of ANP32 is involved in NP recruitment during cRNP and vRNP assembly [9]. The requirement of newly synthesized viral proteins in addition to host factors in replication means that while transcription occurs early in an infection, genome replication is only detectable later, after sufficient levels of viral proteins accumulate.

Much of our current molecular understanding on vRNP functions has been gained through experiments performed in the context of cells or cellular extracts. Dozens of host factors that regulate vRNP functions have been identified using assays such as pull-downs of viral polymerase from transfected or infected cells followed by the identification of interacting host proteins by mass spectrometry [6,10]. However, the precise role of each viral and host factor remains unclear due to the complexity of the cellular environment in which they were studied. Here, we successfully establish a robust in vitro system that reconstitutes influenza virus transcription and, importantly, a full cycle of genome replication, in a test tube. Through this system, we demonstrate the minimal protein requirements for a complete viral genome replication cycle. Additionally, we reveal new roles for ANP32 proteins and NP that are involved in replication. Our system presents a powerful tool for elucidating the roles of regulatory factors in vRNP functions and paves the way for imaging viral replication using cryogenic electron microscopy techniques. Furthermore, our method could be implemented to investigate the molecular mechanisms of genome replication in other negative-sense RNA viruses.

## Results

### Viral polymerase and human ANP32B represent the minimal components to switch the activity of vRNP from transcription to replication

To address the role of viral and host factors in the regulation of influenza virus transcription and replication, we purified virion-derived vRNP of influenza A/WSN/33 (H1N1) virus produced by infecting Madin-Darby bovine kidney (MDBK) cells and using glycerol gradient centrifugation, as previously described (S1A Fig) [11]. The molarity of purified vRNP was estimated by comparing it to a serial dilution of recombinant viral polymerase of known concentrations (S1B Fig). To study the minimal requirements of additional factors in influenza virus genome replication, we purified recombinant viral polymerase expressed in insect cells, as well as ANP32 proteins expressed in *E. coli* (S2 Fig). Technical difficulties were encountered during the expression and purification of recombinant A/WSN/33 (H1N1) polymerase; therefore, we utilized a recombinant A/NT/60/1968 (H3N2) polymerase, which shares over 96% sequence identity and nearly 99% sequence similarity with its homologue from A/WSN/33 (H1N1), for subsequent experiments. We also purified a mutant polymerase with a double

amino acid mutation in its polymerase active site (D445A/D446A, PB1a) rendering it incapable of synthesising RNA (S2A Fig).

To evaluate the role of these viral and host factors, we set up an in vitro vRNP replication reconstitution assay, followed by the analysis of RNAs by primer extension. To prime the transcriptional activity of the vRNP-resident polymerase, globin mRNA was included in the assay as donor of capped RNA primers. In the absence of ribonucleotide triphosphate (rNTP) substrates, only vRNA could be detected, representing the vRNA derived from the purified vRNPs (Fig 1A, lane 1). In the presence of rNTPs, a major mRNA and a minor cRNA product was observed (lane 2), as reported previously [11]. The addition of ANP32B alone did not affect RNA synthesis (lane 3), while the addition of viral polymerase alone, independent of its RNA synthesis activity, resulted in the inhibition of mRNA synthesis (lanes 4 and 6). However, the addition of ANP32B and polymerase together resulted in a robust increase in cRNA synthesis (lanes 5 and 7). This robust increase was observed independent of whether active (wild type (WT)) or inactive (PB1a) polymerase was added (compare lanes 4 and 5 to lanes 6 and 7), demonstrating that replication was carried out by the vRNP-resident polymerase and the role of the additional polymerase is noncatalytic. Therefore, the additional polymerase likely promotes the replicational activity of the vRNP-resident polymerase in an allosteric manner. Similar results were obtained in a set of assays where capped RNA was omitted, apart from no mRNA being detected (Fig 1B).

To test the effect of these factors in a dose-dependent manner, we titrated increasing amounts of viral polymerase and/or ANP32B into the vRNP replication reconstitution assay in the presence of globin mRNA as a cap donor. We observed a gradual decline in mRNA before its complete disappearance and a concomitant increase in the accumulation of cRNA (Fig 1C). In contrast, titration of ANP32B alone had no effect (Fig 1D), while titration of viral polymerase alone resulted in a decline in mRNA levels (Fig 1E). These results suggest that free polymerase alone can inhibit the transcriptional activity of vRNP-associated polymerase; however, for its ability to promote the replicase activity, ANP32 is essential. Taken together, our data demonstrate that non-template-bound viral polymerase and host ANP32 proteins act together to switch the activity of vRNPs from transcription to replication (Fig 1F).

## ANP32 isoforms support replication to different levels, and LCAR is required for replication

Species-specific differences in ANP32 family proteins underpin the poor replication of avian influenza A viruses in mammalian hosts [12,13]. Specifically, mammalian ANP32 proteins cannot support the function of avian influenza A virus polymerase. To further investigate the role of ANP32 family proteins, we compared the ability of all 3 known human isoforms of ANP32 proteins, ANP32A, ANP32B, and ANP32E [14], to support viral genome replication in the presence of viral polymerase (Figs 2A and S2B). We also included chicken ANP32A, as a representative of an avian ANP32, which contains an insertion of 33 amino acids in its LCAR compared to human ANP32A [12,13] (Figs 2A and S2B). Interestingly, among all ANP32 proteins tested, chicken ANP32A showed the strongest support of replication. This is in agreement with previous reports that chicken ANP32A exhibits particularly strong binding for the viral polymerase [15,16]. Among the human ANP32 proteins, ANP32B showed the strongest support of replication, followed by human ANP32A, while human ANP32E could not, or only very weakly, promote replication. The ability of each ANP32 protein to inhibit mRNA levels in the presence of viral polymerase correlated with their ability to promote replication, with chicken ANP32A being the most potent and human ANP32E showing no additional transcriptional inhibition beyond that of the polymerase alone (Fig 2A). The finding that human

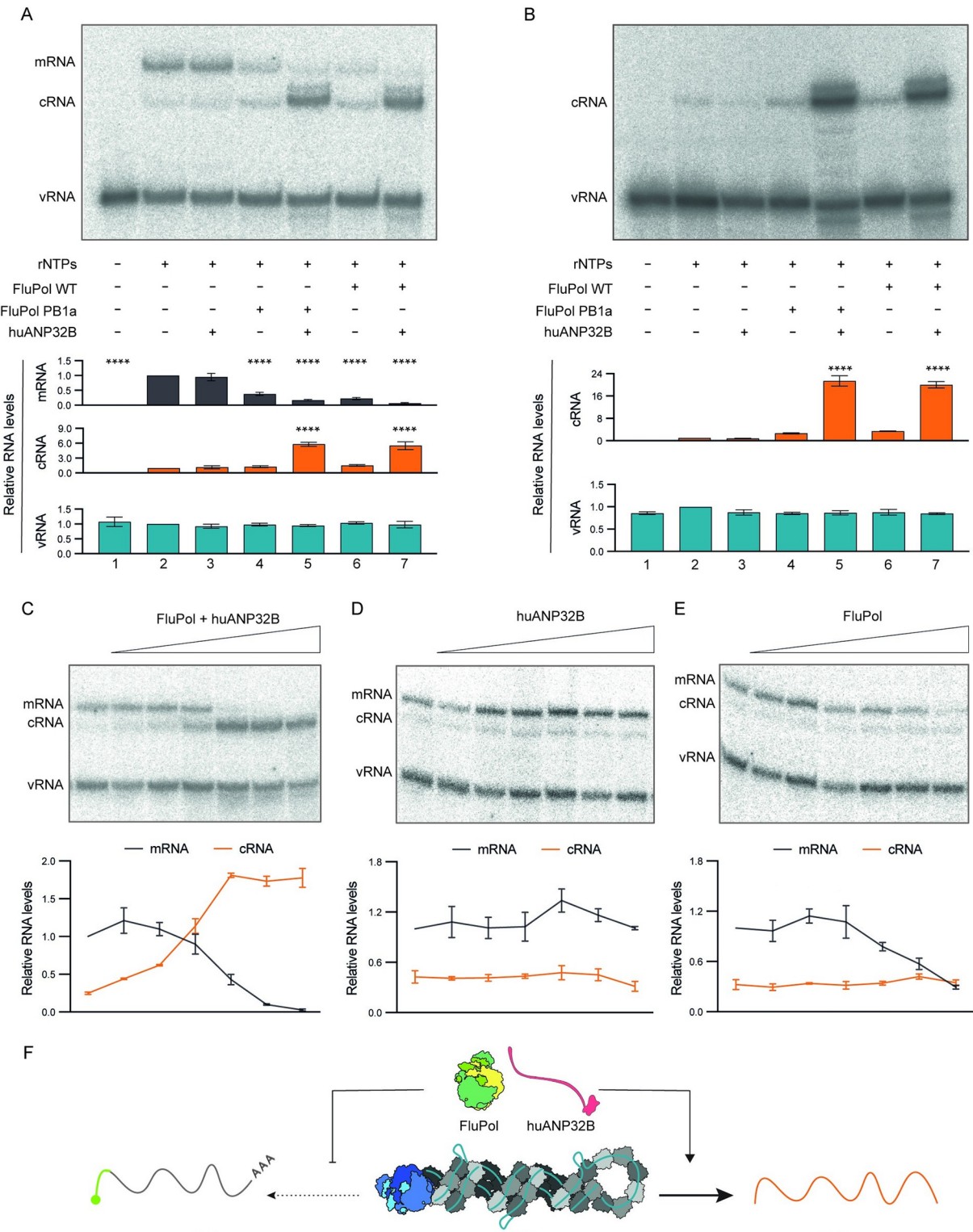

**Fig 1. Viral polymerase (FluPol) and human ANP32B (huANP32B) together promote vRNP replication.** (**A, B**) Effect of active (WT) or inactive (PB1a) FluPol and huANP32B alone or together on vRNP activity, in the presence (**A**) or absence (**B**) of globin mRNA as cap donor. A representative result of RNA analysis by primer extension is shown (top). Relative RNA levels (bottom) were calculated by comparing them to the RNA levels obtained in the absence of FluPol and huANP32B. Ordinary one-way ANOVA was used to assess significance. (**C-E**) Effect of increasing amounts of FluPol and huANP32B (**C**), huANP32B alone (**D**), or FluPol alone (**E**). A representative result of RNA analysis by primer

extension is shown (top). Relative RNA levels (bottom) were calculated by comparing them to the mRNA levels without added FluPol and/or huANP32B. In (**A-E**), data represent mean values ± SEM, based on $n \geq 3$ independent vRNP purifications and reactions. ****, $p < 0.0001$. Original images can be found in S1 Raw Images. Original data sets are in S1 Data. (**F**) Schematic showing the effect of FluPol and huANP32B on the transcriptional and replicational activity of vRNP.

ANP32E is poor at supporting replicational activity of vRNP is in agreement with previous findings using genetically engineered cell lines [17,18].

We then asked whether the C-terminal LCAR is required for the activity of ANP32 in supporting polymerase function. To answer this, we generated a series of chicken ANP32A mutants with truncated LCAR and tested them in the vRNP replication reconstitution assay (S2B Fig). Deletions of up to 46 amino acid residues from the C-terminus of chicken ANP32A (1–250 and 1–235) did not dramatically change vRNP activity when compared with full-length chicken ANP32A (1–281). However, when further amino acid residues were deleted (1–220, 1–188, 1–158, 1–149), the replication promoting ability of ANP32A was diminished (Fig 2B), and no further inhibition of transcriptional activity was observed beyond that with polymerase alone (Fig 2B).

Taken together, these results demonstrate that different members of the ANP32 family proteins have distinct effects on vRNP activity, and the LCAR plays a previously uncharacterised role in promoting genome replication, in addition to NP recruitment [9].

## NP stimulates replicational activity and enables a complete cycle of viral genome replication

We found that while the addition of viral polymerase and ANP32B to vRNPs resulted in robust cRNA synthesis, vRNA levels remained unchanged (Figs 1A–1C, 2A and 2B), indicating that the generated cRNA could not act as template for vRNA synthesis. Previous studies showed that viral NP is an essential factor for replication of the viral genome, as only short templates (<100 nucleotides) could be replicated in its absence [19]. Furthermore, early studies implicated that NP acts as a switching factor between transcription and replication [20,21]. To address the role of NP in replication, we purified NP of influenza A/NT/60/1968 (H3N2) virus expressed in *E. coli* (S3A Fig). Addition of recombinant NP alone to the vRNP replication reconstitution assay, with globin mRNA included as cap donor, had no effect on cRNA levels, indicating that NP alone does not promote the replicational activity of vRNPs (Fig 3A, compare lane 3 to lane 2). Interestingly, we observed a decrease in the mRNA level, suggesting that, similarly to free viral polymerase, NP could act as an allosteric modulator of the vRNP-resident polymerase, inhibiting its transcriptional activity. NP had no additional effect when added in combination with ANP32B or viral polymerase (compare lanes 4 and 5 to lane 2). However, when it was added together with ANP32B and viral polymerase, we observed a substantial increase in cRNA synthesis compared to when only ANP32B and polymerase were added (compare lane 7 to lane 6), independent of whether active (WT) or inactive (PB1a) polymerase was used (compare lanes 7 and 10). Excitingly, when active polymerase was used we also observed a distinct increase in the vRNA level, while no such increase was observed with inactive polymerase (compare lane 10 to lane 7). These data show that NP is an additional significant factor in promoting cRNA synthesis by a vRNP, and product cRNA can act as template for the synthesis of vRNA when active viral polymerase, ANP32B, and NP are present. The latter suggests that including NP in addition to viral polymerase and ANP32 protein enables product cRNA to be assembled into cRNPs that can carry out vRNA synthesis. However, it should be noted that there was only a relatively modest increase in vRNA signals compared to the increase in cRNA signals in response to the addition of polymerase, ANP32B, and NP.

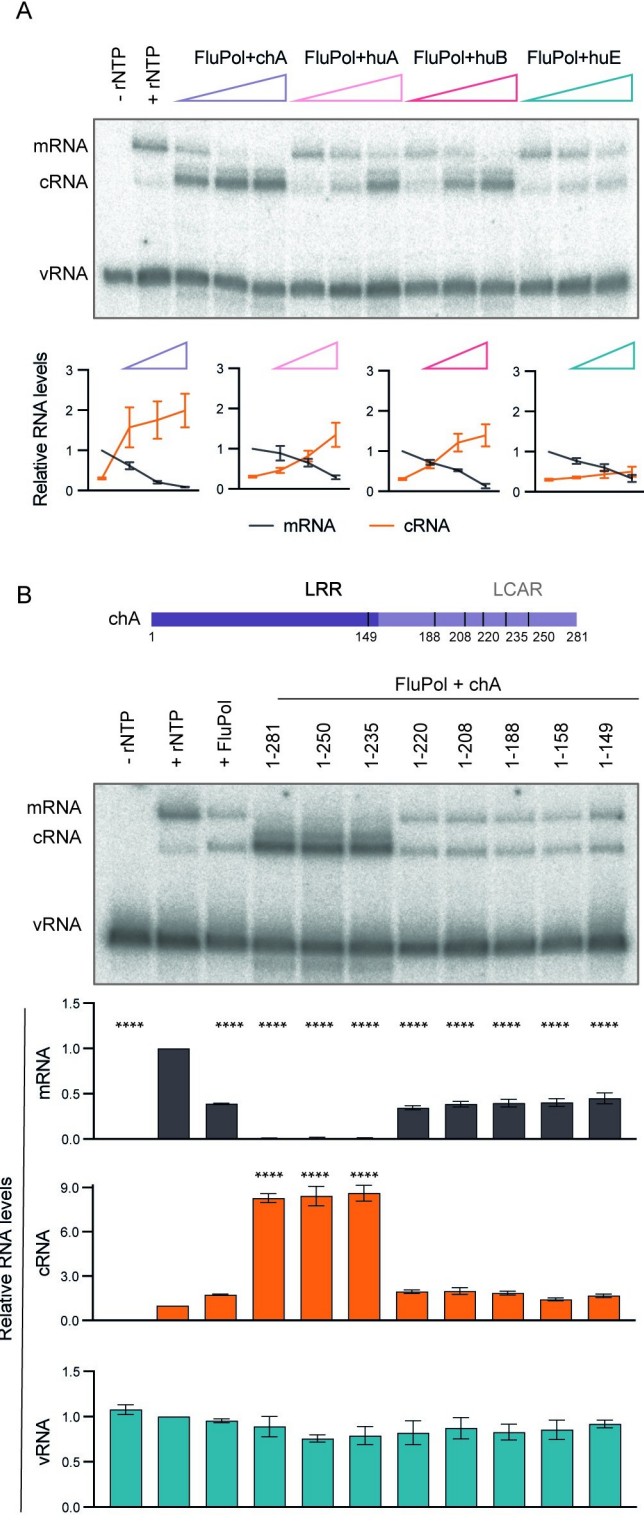

**Fig 2. Effect of ANP32 isoforms and truncated mutants on vRNP replication.** (**A**) Effect of increasing amounts of viral polymerase (FluPol) and chicken ANP32A (chA), human ANP32A (huA), human ANP32B (huB), or human ANP32E (huE) on vRNP activity in the presence of globin mRNA as cap donor. A representative result of RNA analysis by primer extension is shown (top). Relative RNA levels (bottom) were calculated by comparing them to the mRNA levels without added FluPol and ANP32. (**B**) Effect of FluPol and full-length or truncated mutant chA on vRNP activity in the presence of globin mRNA as cap donor. Schematic of chA and truncated mutants lacking parts or the

complete LCAR. LRR and LCAR are shown in dark and light purple, respectively. The numbers under the schematic indicate the length of each mutant in amino acids (top). A representative result of RNA analysis by primer extension is shown (middle). Relative RNA levels (bottom) were calculated by comparing them to the RNA levels obtained in the absence of FluPol and chicken ANP32A. Ordinary one-way ANOVA was used to assess significance. In (**A** and **B**), data represent mean values ± SEM, based on $n \geq 3$ independent vRNP purifications and reactions. ****, $p < 0.0001$. Original images can be found in S1 Raw Images. Original data sets are in S1 Data. ANP32, acidic nuclear phosphoprotein 32; LCAR, low complexity acidic region; LRR, leucine-rich repeat; vRNP, viral ribonucleoprotein.

This observation is consistent with not all cRNA being assembled into functional cRNPs, although the reasons behind this remain unclear. Similar results were observed when globin mRNA was omitted from the vRNP assay (Fig 3B). Titration of increasing amounts of NP into the vRNP replication reconstitution assay in the presence of constant amount of inactive viral polymerase and ANP32B further increased cRNA levels (Fig 3C), while the same setup but using active viral polymerase resulted in a robust, concentration-dependent increase in both cRNA and vRNA levels (Fig 3D), providing further evidence for the roles of NP in genome replication.

NP possesses both oligomerisation and RNA-binding activities [22]. In vRNPs and cRNPs, these 2 properties enable NP to form a polar oligomer by inserting its tail-loop into a groove on the neighbouring molecule and provide a scaffold for the binding of RNA. To analyse the roles of NP in genome replication further, we introduced several mutations into NP and purified the mutant NPs expressed in *E. coli* (S3A Fig). Specifically, we replaced 8 arginine residues with alanine residues in the proposed RNA-binding grooves to generate an RNA-binding mutant (8A), we introduced an R416A mutation into the tail-loop to generate a monomeric mutant (M, defective in oligomerisation) and combined these mutations to produce a monomeric NP that is also deficient in RNA-binding (M8A) [23–26]. We showed previously by size exclusion chromatography that the R416A NP mutant from the A/NT/60/1968 (H3N2) virus behaves as a monomer [27], and here we confirmed that both M and M8A elute as monomers during purification on a Superdex 200 increase 10/300 GL column, while wild type (WT) and the 8A mutant NP elute as oligomers (S3B Fig). By electrophoretic mobility shift assay, we confirmed that the 8A and M8A mutants are deficient in RNA binding (S3C Fig). We then tested the effect of these mutants in the vRNP replication reconstitution assay (Fig 3E). We found that the monomeric mutant NP (M) could promote both cRNA and vRNA synthesis to comparable levels as WT NP (compare lane 4 to lane 3). However, the RNA-binding mutant NP (8A) was no longer able to support a full cycle of viral genome replication, in particular synthesizing vRNA on cRNA templates, but curiously partially retained its ability to promote cRNA synthesis from the input vRNA template (compare lane 5 to lane 3). The monomeric mutant NP, which is also deficient in RNA binding (M8A), exhibited a similar phenotype as 8A, promoting cRNA synthesis but no longer supporting vRNA synthesis (compare lane 6 to lane 3).

Our data show that NP has a function in directly stimulating the replicational activity of the polymerase, independent of its RNA-binding ability. Therefore, NP not only acts as a scaffold for RNA and facilitates its copying by the viral polymerase but also regulates replicational activity. The finding that an oligomerisation-defective mutant NP that retains RNA-binding can still support genome replication, at least in vitro, suggests that oligomerization is not an absolute requirement for NP's role in supporting genome replication. On the other hand, the RNA binding ability of NP is crucial for the assembly of replication competent cRNPs. These results align with previous research indicating that RNA binding, not homo-oligomerization, is essential for NP's ability to support cRNA accumulation in infected cells [28]. However, it should be noted that monomeric NP supports cRNA accumulation at reduced levels compared to WT and NP oligomerisation is required for the replication of full-length genome segments in assays performed in cells [19,28]. We speculate that cRNPs and vRNPs assembled with monomeric NP might have increased susceptibility to degradation by cellular nucleases.

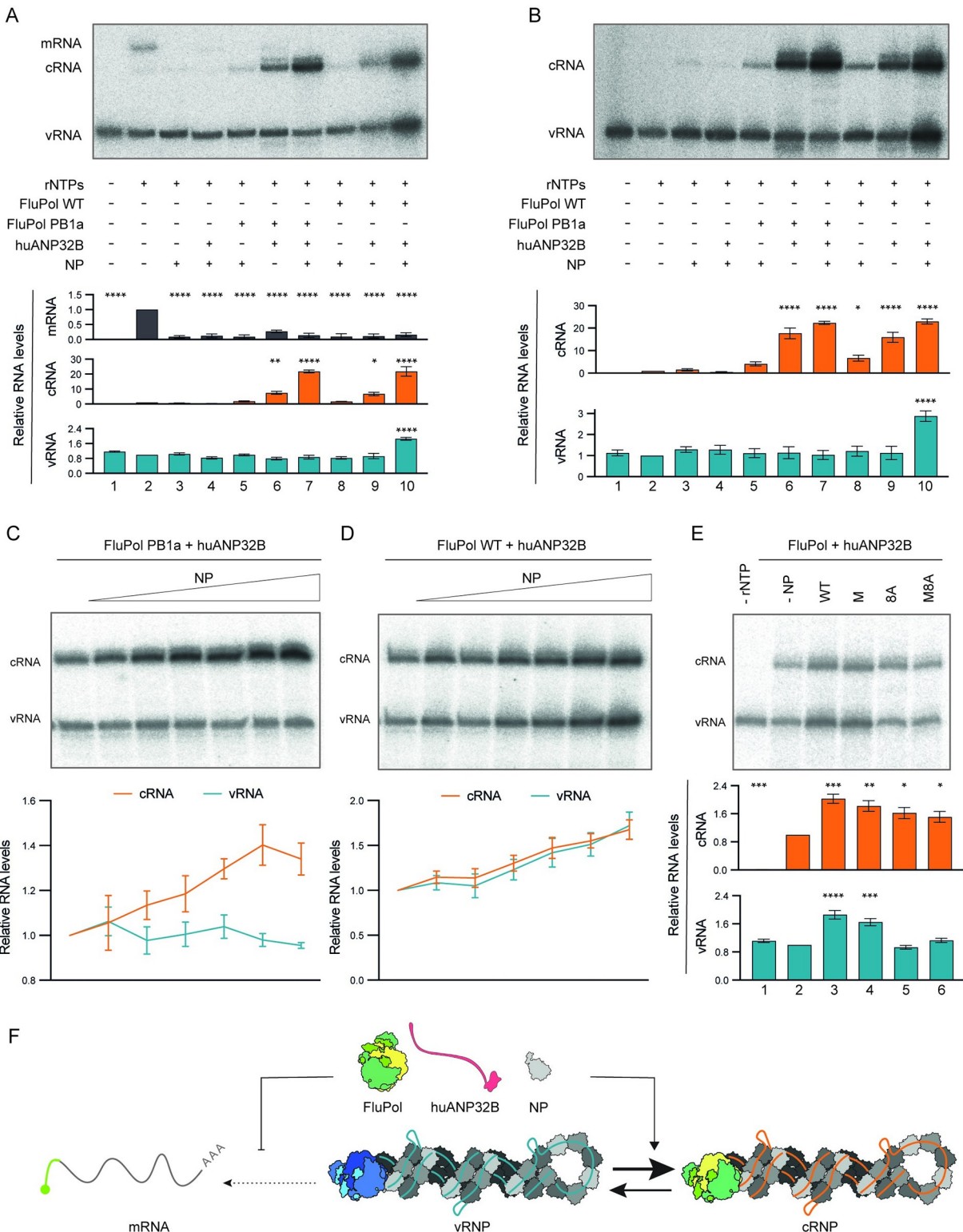

**Fig 3. Viral NP promotes vRNP replication and enables a complete cycle of viral genome replication.** (**A, B**) Effect of NP alone or in combination with inactive (PB1a) or active (WT) viral polymerase (FluPol) and human ANP32B (huANP32B) on vRNP activity, in the presence (**A**) or absence (**B**) of globin mRNA as cap donor. A representative result of RNA analysis by primer extension is shown (top). Relative RNA levels (bottom) were calculated by comparing them to the RNA levels obtained in the absence of FluPol, huANP32B, and NP. Ordinary one-way ANOVA was used to assess significance. (**C, D**) Effect of increasing amounts of NP in the presence of a constant amount of huANP32B and inactive FluPol (PB1a) (**C**) or active FluPol (WT) (**D**) on vRNP activity. A representative result of RNA analysis by primer extension is shown

(top). Relative RNA levels (bottom) were calculated by comparing them to the RNA levels without added NP. (**E**) Effect of oligomerisation capacity and RNA-binding ability on the functions of NP in viral replication. A representative result of RNA analysis by primer extension is shown (top). Relative RNA levels (bottom) were calculated by comparing them to the RNA levels obtained in the absence of NP. Ordinary one-way ANOVA was used to assess significance. In (**A-E**), data represent mean values ± SEM, based on $n \geq 3$ independent vRNP purifications and reactions. *, $p < 0.05$; **, $p < 0.01$; ***, $p < 0.001$; ****, $p < 0.0001$. Original images can be found in S1 Raw Images. Original data sets are in S1 Data. (**F**) Schematic showing that FluPol, huANP32B, and NP are the minimal set of proteins required for a complete cycle of viral genome replication. NP, nucleoprotein; vRNP, viral ribonucleoprotein; WT, wild type.

In summary, our results show that NP plays 2 roles in genome replication: promoting cRNA synthesis on vRNA templates in the presence of viral polymerase and ANP32 proteins, and enabling a full cycle of replication, that is, vRNA to cRNA and cRNA to vRNA synthesis (Fig 3F and S4 Fig). Importantly, by demonstrating that viral polymerase, ANP32 proteins, and NP are the minimal factors required for vRNP to accomplish a full cycle of replication, we establish the first complete in vitro replication reconstitution system for influenza virus.

## Discussion

In this study, we reconstituted a major step of the influenza virus life cycle, viral genome replication, in a test tube. Using vRNPs isolated from purified virions and recombinant viral polymerase, NP, and host ANP32 proteins, we established an entirely cell-free vRNP replication reconstitution assay for the influenza virus RNA genome that faithfully recapitulates both steps of replication, vRNA to cRNA and cRNA to vRNA synthesis, and the assembly of functional cRNPs. We define that viral polymerase and ANP32 proteins are the minimal components required for the replicational activity of a vRNP, while NP further promotes replicational activity and is essential for the assembly of functional cRNPs that can carry out vRNA synthesis. Presumably, vRNA also assembles into functional vRNPs that can subsequently carry out further cRNA synthesis. However, this remains to be determined, as our current assay cannot distinguish cRNA produced by in vitro assembled vRNPs from that synthesised by input vRNPs.

Viral polymerase switches between different enzymatic activities to support transcription and replication at different stages of viral infection. We found that vRNPs isolated from virions preferentially synthesize mRNA through the vRNP-resident polymerase, which acts as a transcriptase. However, the addition of viral polymerase and ANP32 proteins converts the transcriptase into a replicase, most likely through viral polymerase and ANP32 protein binding to the vRNP-resident polymerase and inducing a conformational rearrangement. These findings are consistent with a recently proposed model for influenza virus genome replication, based on a cryo-EM structure of an influenza C virus polymerase dimer bound to an ANP32A protein [8]. According to this model, newly translated viral polymerase is targeted to the resident polymerase of a transcribing vRNP to form a dimer, which is mediated by the N-terminal LRR domain of an ANP32A protein. The vRNP-resident polymerase acts as a replicase, while the additional polymerase acts as an encapsidating polymerase that captures the 5′ end of the nascent cRNA and enables the co-replicational assembly of product cRNA into a cRNP. The highly acidic C-terminal LCAR of ANP32A has been proposed to act as a platform for increasing local NP concentration around the replicating vRNP and promoting NP recruitment to nascent cRNA during the assembly of cRNPs [9]. These newly assembled cRNPs then give rise to vRNA synthesis, with additional viral polymerase and ANP32 proteins activating the replicational activity of the cRNP-resident polymerase. Our work provides evidence for this model by showing that non-vRNP-associated polymerase, ANP32, and NP are the minimal components required for a full cycle of replication of a vRNA genome segment in the context of a vRNP and offers further insights into the workings of this molecular machine (Fig 4).

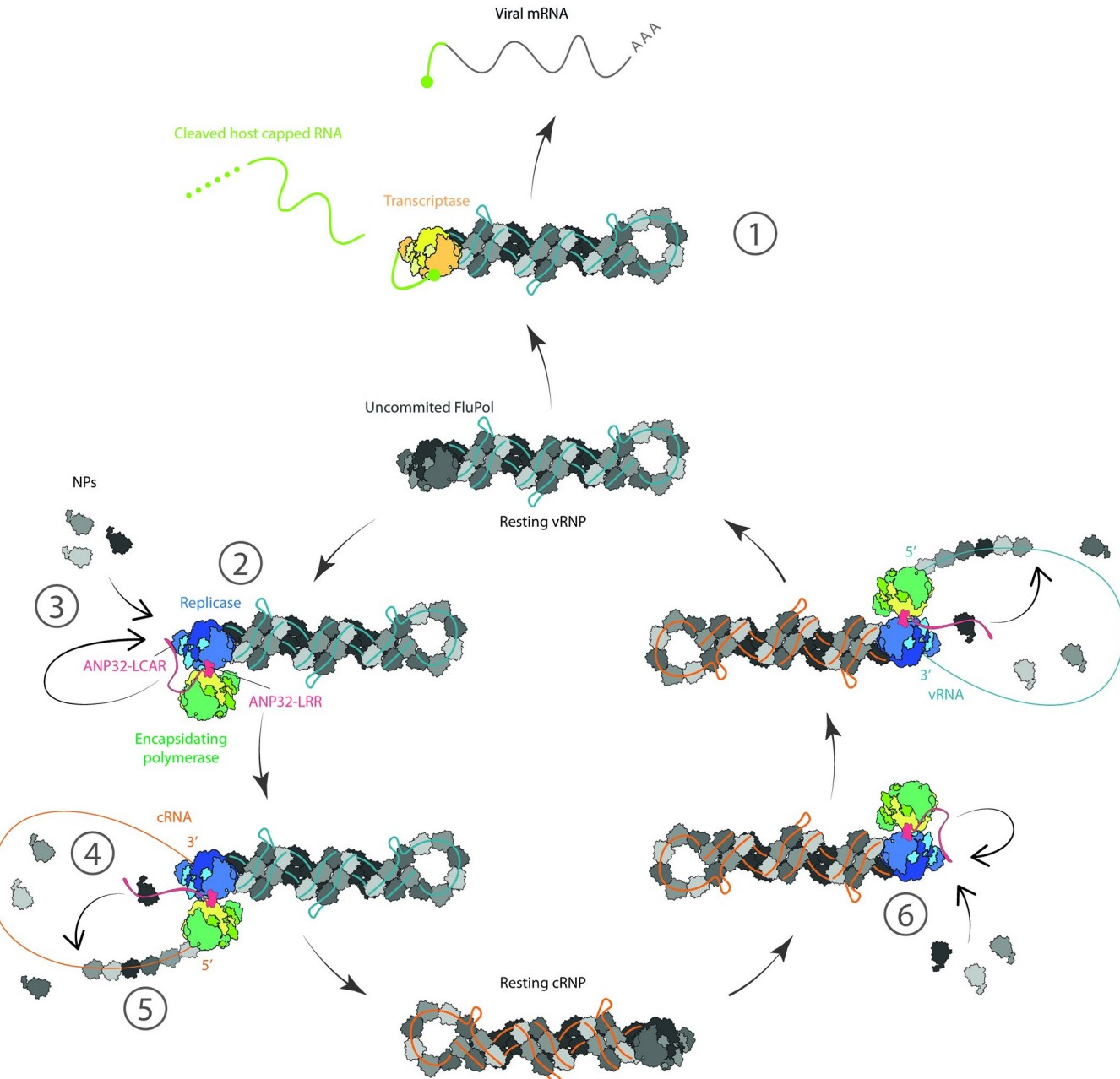

**Fig 4. Model summarising the modulation of vRNP activity by viral and host factors in the in vitro vRNP replication reconstitution assay.** Virion-derived vRNPs preferentially synthesize mRNA in the presence of a capped RNA primer donor with the vRNP-resident polymerase acting as transcriptase (1). Additional viral polymerase and ANP32 bind to the vRNP-resident polymerase and switch the vRNP-resident polymerase, by allosteric modification, from a transcriptase to a replicase that initiates cRNA synthesis in a primer-independent manner (2). The ANP32-LRR stabilises the polymerase dimer, while the ANP32-LCAR plays a role in promoting genome replication through a yet to be discovered mechanism. NP further promotes replication in an RNA-binding independent manner (3). The 5′ end of cRNA product is captured by the encapsidating polymerase, and the ANP32-LCAR promotes the co-replicational assembly of cRNA into cRNP by binding to NP and increasing its local density (4). The RNA-binding ability of NP is required for the assembly of functional cRNP that can carry out the second step of replication (5). Additional polymerase and ANP32 bind to the resident polymerase of newly assembled cRNP, converting the uncommitted polymerase to a replicase to perform primer-independent initiation of vRNA synthesis (6). The synthesis of vRNA and its assembly into vRNP proceed in a manner similar to that described for cRNA above. ANP32, acidic nuclear phosphoprotein 32; cRNA, complementary RNA; LCAR, low complexity acidic region; LRR, leucine-rich repeat; NP, nucleoprotein; vRNP, viral ribonucleoprotein.

Non-template-bound, free viral polymerase has been shown to be important for the accumulation of cRNA replication products in virus-infected cells and moderately promoted the activity of isolated cRNPs in vitro [29,30]. Interestingly, a catalytically inactive form of viral polymerase was also able to stimulate activity, suggesting that it is the RNP-resident polymerase that acts as replicase while the second viral polymerase acts as an encapsidating polymerase [29,30], challenging an earlier model in which polymerase was proposed to act in *trans*, being responsible for the synthesis of the replication product [31]. In this study, we firmly established that it is the vRNP-resident polymerase that acts as replicase while the additional viral polymerase performs a noncatalytic allosteric role. We also confirm that ANP32 family proteins, being an essential part of the replication machinery, play an indispensable role in supporting influenza viral genome replication. Interestingly, we found that avian and human ANP32 isoforms differ in their potential to promote genome replication, with avian (chicken) ANP32A having the strongest effect together with viral polymerase, leading to a complete suppression of transcriptional activity and a strong promotion of cRNA synthesis. This could be related to the increased affinity of avian ANP32A for viral polymerase as reported previously [15,16]. Interestingly, we also found that most of the LCAR is required for supporting cRNA synthesis, independent of the presence of NP, strongly suggesting that the LCAR is involved in another function in addition to NP recruitment, most likely directly interacting with the viral polymerase.

The role of NP in regulating vRNP function has been a subject of debate for many years. Early studies suggested that NP is the critical factor that switches the polymerase from transcription to replication [20,21]. However, subsequent studies demonstrated that NP stabilises replication products in cells, along with viral polymerase, by virtue of its RNA-binding ability, and plays no role in regulating polymerase function [30]. Here, we present evidence that NP is required for a complete cycle of viral genome replication. We also identified a new role of NP in direct replication promotion that is independent of its RNA-binding ability. By isolating the 2 steps of influenza virus replication, we propose that NP can act as an additional replication cue, presumably by interacting with the replication platform to further promote replication. This could be advantageous as NP is abundant, and binding of NP to the replicating platform could create a pool of NP available for recruitment by ANP32 family proteins in RNP assembly. Overall, our results provide insight into the versatile roles of NP in influenza virus replication and offer new directions for further investigation.

The dependency of the vRNP-resident polymerase on the presence of free viral polymerase for its replicational activity, along with the stimulatory role of NP, provides influenza virus with an elegant mechanism to time its genome replication such that it only occurs when sufficient levels of polymerase and NP accumulated in the infected cell. This timing ensures that no naked cRNA and vRNA are produced, which, possessing 5′ triphosphates and a double-stranded panhandle region as a result of the partial complementarity of the termini, could be detected by pathogen recognition receptors such as RIG-I to trigger the activation of innate immune pathways and suppress influenza virus replication [32]. Nevertheless, influenza virus infection is known to induce innate immune activation and defective viral RNAs have been implicated as primary innate immune inducers [33]. Our system could be suitable for assessing the fidelity of the RNA polymerase and its potential to generate defective viral RNAs and as such could provide a platform for evaluating the role of the influenza virus RNA polymerase in innate immune activation and, consequently, as a determinant of virulence.

We show in this study that influenza virus genome replication is dependent on free non-vRNP associated viral polymerase and an ANP32 protein; nevertheless, we also show that vRNPs can perform low level of cRNA synthesis in the absence of any additional factor. The question arises whether this low-level activity is simply an in vitro artefact or could be

biologically relevant. If cRNA were produced early in infection when no or only very low levels of free viral polymerase and NP are present, such cRNA would be quickly degraded by host nucleases. However, we cannot exclude the possibility that some cRNA could escape degradation and being detected by RNA sensors could trigger an innate immune response. It would be interesting to explore whether different influenza A virus strains, that is, human seasonal, human pandemic, and avian, might be able to produce different levels of cRNA early on and therefore induce different levels of innate immune response that could impact virulence.

In summary, we have, for the first time, reconstituted a full cycle of influenza virus genome replication in a test tube. This robust assay provides a highly controlled and simplified system to understand the molecular details of influenza virus replication. Using this novel in vitro system, we defined the minimal requirement of viral and host factors that are essential for viral genome replication and showed intriguing new roles of viral and host factors in this process. This breakthrough opens up avenues for further exploration of influenza virus genome replication, including the use of cryo-electron microscopy techniques to observe vRNPs at different stages during replication. Additionally, this methodology could be expanded to investigate other negative-sense RNA viruses.

## Materials and methods

### Cell lines

MDBK cells were sourced from the Cell Bank of the Sir William Dunn School of Pathology. MDBK cells were cultured in minimal essential medium (MEM, Gibco) supplemented with 10% FBS and 2 mM L-glutamine (Gibco) at 37°C with 5% $CO_2$. *Spodoptera frugiperda* Sf9 cells were kind gift from Dr. Weixian Lu (Division of Structural Biology, University of Oxford) and maintained in Sf-900 II serum-free medium at 27°C shaking at 110 revolutions per minute (rpm). Cell lines have not been authenticated but tested negative for mycoplasma contamination.

### Viral infections

MDBK cells at a confluency of 80% were infected with influenza A/WSN/33 (H1N1) virus at a multiplicity of infection (MOI) of 0.01 in MEM containing 0.5% foetal bovine serum (FBS) and 2 mM L-glutamine. At 48 h postinfection, cell culture medium containing virus was collected and cell debris was removed by centrifugation at 2,000$g$ for 10 min at 4°C, followed by 17,000$g$ for 15 min at 4°C.

### Virion-derived vRNP purification

Viron-derived vRNP was prepared essentially as described previously [11]. Briefly, cell culture medium from infected cells was loaded onto precooled 30% sucrose cushion (30% sucrose, 100 mM NaCl, 10 mM Tris-HCl (pH 7.4), 1 mM EDTA) in thick-walled ultracentrifuge tubes. Virus was pelleted by centrifugation at 105,000$g$ for 1.5 h at 4°C and resuspended in chilled resuspension buffer (100 mM NaCl, 10 mM Tris-HCl (pH 7.4), 1 mM EDTA) at 4°C. Resuspended virus was lysed by incubation in a disruption buffer (100 mM Tris-HCl (pH 7.4), 100 mM NaCl, 5 mM MgCl$_2$, 1% Triton X-100, 5% glycerol, 0.5% Igepal, freshly added 20 mg/ml lysophosphatidylcholine (Sigma-Aldrich) and 1.5 mM DTT) at 31°C for 30 min with vigorous shaking. The viral lysate was fractionated by centrifugation on a discontinuous glycerol gradient (1 ml each 70%, 50%, 40%, and 33% glycerol in 50 mM Tris-HCl (pH 7.4) and 150 mM NaCl). The gradient was centrifuged at 250,000$g$ in a Beckman SW55Ti rotor for 4 h at 4°C. Fractions of approximately 250 μl were collected dropwise from the bottom of the tube.

Proteins were analysed by sodium dodecyl sulfate polyacrylamide gel electrophoresis (SDS-PAGE) and silver stained. Around 3 to 4 fractions enriched with vRNP were pooled and pelleted in an Optimax-XP TLA 100.3 fixed angle rotor in thick-walled polycarbonate tubes at 550,000g for 4 h at 4˚C. vRNP pellet was resuspended in vRNP buffer (100 mM HEPES-NaOH (pH 8.0), 150 mM NaCl, 10% glycerol, 1× phenylmethylsulfonyl fluoride (PMSF)), aliquoted, snap-frozen, and stored at −80˚C. The molarity of purified vRNP was estimated by comparing it to a serial dilution of recombinant viral polymerase of known concentration (S1B Fig).

## Plasmids for recombinant protein production

The genes of the 3 subunits of influenza A/NT/60/1968 (H3N2) virus polymerase were codon optimized and synthesized by ThermoFisher and cloned into a single baculovirus using the MultiBac system [34]. Plasmid expressing the active site mutant polymerase (D445A/D446A double mutation in the PB1 subunit, PB1a [30]) was created by site-directed mutagenesis. WT and mutant NP from influenza A/NT/60/1968 (H3N2) virus, as well as full-length and truncation mutant ANP32 family proteins, were cloned into pGEX-6P-1 vector (GE Healthcare) with an N-terminal GST tag followed by a PreScission protease site. Mutant NP plasmids (monomeric: R416A, M; RNA-binding deficient: R74A/R75A/R150A/R156A/R160A/R174A/R175A, 8A; and monomeric and RNA-binding deficient mutant: the combination of the previous two, M8A [23–26]) were generated by site-directed mutagenesis. Plasmids encoding truncated chicken ANP32A 1–149, 1–158, 1–188, 1–208, 1–220, 1–235, and 1–250 were also generated by site-directed mutagenesis.

## Recombinant protein expression and purification

Sf9 insect cells were infected with recombinant baculovirus expressing the subunits of the influenza A/NT/60/1968 (H3N2) virus polymerase (WT or PB1a mutant) and harvested 72 h postinfection. The purification of polymerase was performed as previously described [35]. Briefly, cell pellet was lysed by sonication and clarified lysate was incubated with IgG sepharose (Cytiva). After several washes and overnight cleavage by TEV protease, the released protein was concentrated and applied on to a Superdex 200 increase 10/300 GL column (Cytiva). Fractions with targeted protein were pooled, concentrated to desired concentration, aliquoted, snap-frozen in liquid nitrogen, and stored at −80˚C until further use. WT and mutant NP from influenza A/NT/60/1968 (H3N2) virus, as well as full-length and truncation mutant ANP32 family proteins, were expressed in BL21 (DE3) *E. coli* strain and harvested 18 h after being induced by 0.5 mM isopropyl β- d-1-thiogalactopyranoside (IPTG) at 18˚C. All these proteins were purified as previously described [9]. Briefly, cell pellet was lysed by sonication and clarified lysate was incubated with GST sepharose (Cytiva). After several washes and overnight cleavage by PreScission protease, the released protein was concentrated and applied on to a Superdex 200 increase 10/300 GL column (Cytiva). Fractions with targeted protein were pooled, concentrated to desired concentration, aliquoted, snap-frozen in liquid nitrogen, and stored at −80˚C until further use.

## In vitro vRNP activity assay

Approximately 100 ng of recombinant vRNP and indicated recombinant proteins diluted in vRNP buffer (100 mM HEPES-NaOH (pH 8.0), 150 mM NaCl, 10% glycerol, 1X PMSF) were mixed with 40 ng rabbit globin mRNA (Sigma-Aldrich) as primer donor (in primed reaction), 1 mM ATP, 0.5 mM GTP, 0.5 mM CTP, 0.5 mM UTP, 5 mM MgCl$_2$, 1 mM DTT, and 2 U μl$^{-1}$ RNasin (Promega) in a reaction volume of 20 μl. After 4-h incubation at 30˚C, RNA was extracted using TRI reagent (Sigma-Aldrich) according to the manufacturer's instructions.

Approximately 1,000 ng of recombinant polymerase (WT and PB1a), 130 ng of recombinant ANP32B, and 800 ng of recombinant NP (WT, M, 8A, and M8A) were added when indicated. In titration experiments, 50 ng, 100 ng, 200 ng, 500 ng, 1,000 ng, and 2,000 ng of FluPol; 6.5 ng, 13 ng, 26 ng, 65 ng, 130 ng, and 260 ng of ANP32B; and 40 ng, 80 ng, 200 ng, 300 ng, 400 ng, and 800 ng of NP were used as indicated. In ANP32 family protein titration experiment, 26 ng, 65 ng, and 130 ng of chANP32A; 23 ng, 57 ng, and 114 ng of human huANP32A/B; and 24.5 ng, 61 ng, and 122 ng of huANP32E were used. In ANP32 protein truncation experiment, 116 ng of 1–250, 108 ng of 1–235, 102 ng of 1–220, 96 ng of 1–208, 86 ng of 1–188, 72 ng of 1–158, and 68 ng of 1–149 were used (equal mol).

## RNA analysis by primer extension

Total RNA was extracted using TRI reagent (Sigma-Aldrich) according to the manufacturer's instructions and subjected to primer extension analysis as previously described [36]. In brief, RNA was reverse transcribed using Super-Script III reverse transcriptase (Invitrogen) with an excess of $^{32}$P-labelled primers specific to segment 6 (encoding the neuraminidase (NA)) positive-sense mRNA and cRNA (5′–TCCAGTATGGTTTTGATTTCCG–3′) or vRNA (5′–TGGAC TAGTGGGAGCATCAT–3′), segment 1 (encoding the polymerase basic 2 protein (PB2)) positive-sense mRNA and cRNA (5′–GCCATCATCCATTTCATCCT–3′) or vRNA (5′–TGCTAA TTGGGCAAGGAGAC–3′), segment 2 (encoding the polymerase basic 1 protein (PB1)) positive-sense mRNA and cRNA (5′–TCCATGGTGTATCCTGTTCC–3′) or vRNA (5′–TGATTTC GAATCTGGAAGGA–3′), segment 5 (encoding the NP) positive-sense mRNA and cRNA (5′– TGATTTCAGTGGCATTCTGG–3′) or vRNA (5′–TGATGGAAAGTGCAAGACCA–3′), segment 8 (encoding the nonstructural (NS) proteins) positive-sense mRNA and cRNA (5′–CGCTCC ACTATTTGCTTTCC–3′) or vRNA (5′–TGATTGAAGAAGTGAGACACAG–3′). Primer extension products derived from mRNA and cRNA can be distinguished based on size because of the presence of globin mRNA-derived sequences at 5′ end of mRNA. Primer extension products were separated by 6% w/v denaturing PAGE and visualised by phosphorimaging on an FLA-5000 scanner (Fuji). Quantitation was carried out using ImageJ (Fiji) and analysis was carried out using Prism 8 (GraphPad). Experiments were performed with $n = 3$ independently purified vRNP preparations. Statistical tests (ordinary one-way ANOVA) were carried out using Prism 8.

## Electrophoretic mobility shift assay (EMSA)

To test the RNA-binding ability of WT and mutant NP, 140 pmol of a 39 nucleotide-long RNA (5′–AGUAGAAACAAGGCCGUAGAAUGAUGUAUAUGAGACAGA–3′, Dharmacon) was mixed with 62, 116, or 232 pmol of WT or mutant NP in binding buffer (25 mM HEPES-- NaOH (pH 7.5), 150 mM NaCl, 5% Glycerol, 2 mM DTT, 2 U μl$^{-1}$ RNAsin (Promega)) in a final volume of 15.5 μl and incubated on ice for 30 min before being analysed on a 10% native polyacrylamide gel. The gel was stained by SYBR Safe (ThermoFisher) to visualize RNA. The experiment was repeated 3 times.

## Supporting information

**S1 Fig. Purification and semi-quantification of vRNP.** (**A**) Schematic illustrating the glycerol gradient purification of vRNPs isolated from virions produced in MDBK cells. (**B**) Estimation of vRNP molar concentration through comparison of vRNP with known concentration of viral polymerase (FluPol). Original image can be found in S1 Raw Images.
(TIF)

**S2 Fig. Analysis of purified recombinant viral polymerase (FluPol) and ANP32 family proteins by SDS-PAGE and staining with Coomassie Brilliant Blue.** (**A**) WT and catalytically inactive (PB1a) FluPol purified from insect cells. (**B**) ANP32 family proteins including chicken ANP32A (chANP32A), human ANP32A (huANP32A), human ANP32B (huANP32B), human ANP32E (huANP32E), and C-terminally truncated chANP32A mutants purified from *E. coli*. * indicates residual cleaved GST tag. Original images can be found in S1 Raw Images. ANP32, acidic nuclear phosphoprotein 32; SDS-PAGE, sodium dodecyl sulfate polyacrylamide gel electrophoresis; WT, wild type.
(TIF)

**S3 Fig. Analysis of WT and mutant viral NP by SDS-PAGE and assessing their RNA-binding activity by EMSA.** (**A**) WT, RNA-binding deficient mutant (8A), monomeric mutant (M), and RNA-binding deficient monomeric mutant (M8A) NP proteins were purified from *E. coli*. * indicates NP degradation product. (**B**) Size-exclusion chromatography curves of the purification of NP proteins. Size markers are shown above the curves. The y-axes were rescaled to make main peaks of equal heights. (**C**) RNA-binding activity of WT and mutant NP assessed by EMSA. Original images can be found in S1 Raw Images. EMSA, electrophoretic mobility shift assay; NP, nucleoprotein; SDS-PAGE, sodium dodecyl sulfate polyacrylamide gel electrophoresis; WT, wild type.
(TIF)

**S4 Fig. Analysis of different influenzas A virus segments.** Effect of viral polymerase (FluPol), huANP32B, and NP alone or together on vRNP activity in segment 1 (PB2), segment 2 (PB1), segment 5 (NP), and segment 8 (NS), in the presence of globin mRNA as cap donor. A representative result of RNA analysis by primer extension is shown. Original images can be found in S1 Raw Images. NP, nucleoprotein; NS, nonstructural; PB1, polymerase basic 1; PB2, polymerase basic 2; vRNP, viral ribonucleoprotein.
(TIF)

**S1 Data. Excel files containing all the raw data for Figs 1, 2 and 3.**
(XLSX)

**S1 Raw Images. Raw images for Figs 1, 2, 3, S1, S2, S3 and S4.**
(PDF)

## Acknowledgments

We thank Imre Berger (University of Bristol) for the Multibac recombinant baculovirus/insect cell expression system and Weixian Lu (University of Oxford) of the Sf9 insect cells. We also thank members of the Fodor and Grimes laboratories for helpful discussions and, specifically, Jeremy Keown for encouraging the initiation of this project, Alex Walker for generating preliminary data, and Jane Sharps for assistance with culturing and purifying influenza virus and the preparation of vRNP.

## Author Contributions

**Conceptualization:** Zihan Zhu, Haitian Fan, Ervin Fodor.

**Formal analysis:** Zihan Zhu.

**Funding acquisition:** Ervin Fodor.

**Investigation:** Zihan Zhu, Haitian Fan.

**Resources:** Ervin Fodor.

**Supervision:** Haitian Fan, Ervin Fodor.

**Writing – original draft:** Zihan Zhu, Haitian Fan.

**Writing – review & editing:** Zihan Zhu, Haitian Fan, Ervin Fodor.

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
