## [Editor Report · Decision Letter 0]

3 Jul 2023

Dear Dr. Fodor, 

Thank you for submitting your manuscript entitled "Defining the minimal components of the influenza A virus replication machinery via an in vitro reconstitution system" for consideration as a Research Article by PLOS Biology.

Your manuscript has now been evaluated by the PLOS Biology editorial staff and I am writing to let you know that we would like to send your submission out for external peer review.

Once your full submission is complete, your paper will undergo a series of checks in preparation for peer review. After your manuscript has passed the checks it will be sent out for review. To provide the metadata for your submission, please Login to Editorial Manager (https://www.editorialmanager.com/pbiology) within two working days, i.e. by Jul 05 2023 11:59PM.

Kind regards,

Paula

---

Senior Editor

PLOS Biology

---

## [Decision Letter · Decision Letter 1]

23 Sep 2023

Dear Dr Fodor,

Thank you for your patience while your manuscript "Defining the minimal components of the influenza A virus replication machinery via an in vitro reconstitution system" went through peer-review at PLOS Biology. Your manuscript has now been evaluated by the PLOS Biology editors, an Academic Editor with relevant expertise, and by several independent reviewers.

In light of the reviews, which you will find at the end of this email, we are pleased to offer you the opportunity to address the comments from the reviewers in a revision that we anticipate should not take you very long. We will then assess your revised manuscript and your response to the reviewers' comments with our Academic Editor aiming to avoid further rounds of peer-review.

Please also address the following policy and formatting requests:

1. DATA POLICY:

A) Supplementary files (e.g., excel). Please ensure that all data files are uploaded as 'Supporting Information' and are invariably referred to (in the manuscript, figure legends, and the Description field when uploading your files) using the following format verbatim: S1 Data, S2 Data, etc. Multiple panels of a single or even several figures can be included as multiple sheets in one excel file that is saved using exactly the following convention: S1_Data.xlsx (using an underscore).

B) Deposition in a publicly available repository. Please also provide the accession code or a reviewer link so that we may view your data before publication. 

Regardless of the method selected, please ensure that you provide the individual numerical values that underlie the summary data displayed in the following figure panels as they are essential for readers to assess your analysis and to reproduce it: Figures 1ABCDE, 2AB, 3ABCDE.

For manuscripts submitted on or after 1st July 2019, we require the original, uncropped and minimally adjusted images supporting all blot and gel results reported in an article's figures or Supporting Information files. We will require these files before a manuscript can be accepted so please prepare and upload them now. We require this for figures 1ABCDE, 2AB, 3ABCDE, and supplementary figures S1B, S2AB, S3AB.

Please carefully read our guidelines for how to prepare and upload this data: https://journals.plos.org/plosbiology/s/figures#loc-blot-and-gel-reporting-requirements

**IMPORTANT - SUBMITTING YOUR REVISION**

*Resubmission Checklist*

*Published Peer Review*

*PLOS Data Policy*

*Blot and Gel Data Policy*

Sincerely,

Paula

---

Senior Editor

PLOS Biology

REVIEWS:

Reviewer #1: RNA virus replication.

Reviewer #2: Influenza.

Reviewer #1: Review on manuscript PBIOLOGY-D-23-01602R1

The authors established an authentic in vitro transcription and replication system for influenza A virus, based on vRNPs isolated from virions and using recombinant viral and host factors essential for these processes. This is an substantial advancement in the research field of negative sense RNA viruses. The authors have defined the minimal components apart from vRNPs, including the host protein ANP32 as well as exogenous viral RNAP and NP. Several mechanistic issues could be clarified. Their assay provides an important platform to address many mechanistic issues of IAV transcription and replication in the future.

Major comments:

1) The text description of results presented in Fig. 1a, b and Fig. 3a, b and e is aggravated for the reader because the authors refer in the text only to entire part figures without referring to the specific relevant lanes. As a consequence the reader has to collect the corresponding lanes her/himself, which is a bit like 'collecting mushrooms'. For example, at first reading I did not catch the conclusion inferred from Fig. 3e, namely that the M8A variant of NP was still able to somewhat stimulate cRNA synthesis despite oligomerisation and RNA-binding deficiency.

The problem can be solved by numbering the lanes in Fig. 1a, b and Fig. 3a, b and e, and referring meticulously to the respective lanes when describing the individual results in the text.

2) Fig. 1a, b and 3a, b: make the - and + signs below the primer extension gels larger and bolder. 

3) Page 7, lines 205/206: here it is particularly important to number the lanes in Fig. 3e and to guide the reader to the lanes showing stimulation of replication (cRNA synthesis) by NP variants 8A and M8A that lost their RNA-binding ability.

Minor comments:

p. 4:

- line 100: "... substrates, only vRNA ..."

- line 105: "... resulted in a robust increase ..."

- line 109: "... polymerase likely promotes the replicational ..."

p. 5:

- line 112: "... we titrated increasing amounts ..."

- line 117: "... resulted in a decline of mRNA ..."

- line 133: "... among all ANP32 proteins tested, chicken ..."

- line 135: "... exhibits particularly strong binding to the viral ..."

- line 137: "... whilst human ANP32E could not, or only very weakly, promote replication."

- line 140: "... showing almost similar ...

p. 6:

- line 146: Supplementary Fig. 2d

- line 161: "... essential factor for replication of the viral genome, as only ..."

- line 175/176: "... distinct increase in vRNA level, while no such increase was observed with inactive polymerase (Fig. 3a)."

p. 7:

- line 183-185: "... further increased cRNA levels (Fig. 3c), whilst the same setup but using active viral polymerase resulted in ..." 

- line 193: "... into the tail-loop ..."

- line 194: "... monomeric mutant (M, defective in NP oligomerisation), and combined these mutations to produce a monomeric NP that is also deficient in RNA binding (M8A) [23-26]. We confirmed by electrophoretic ..."

- line 196: "... that the 8A and M8A mutants are deficient ..."

- line 201: " on cRNA templates, but curiously ..."

- line 208: " The finding that an oligomerisation-defective mutant NP that retains RNA binding can ..."

p. 8:

- line 212: "... RNA binding ..."

- line 215: "... synthesis on vRNP templates in the presence ..."

- line 216: "... full cycle of replication, that is, vRNA to cRNA and cRNA to vRNA synthesis (Fig. 3f)."

p. 9:

- line 242: "... polymerase that captures the 5' end of the nascent cRNA and enables the ..."

- line 246: " These newly assembled cRNPs then give rise to vRNA synthesis, with ..."

- line 253: "... virus-infected cells ..."

- line 263: "... genome replication, with avian ..."

p. 10:

- line 288: "... double-stranded ..."

- line 290: "... such as RIG-I to trigger the activation of innate immune pathways and supress influenza ..."

- line 302: "... viral polymerase and NP are present, such ..."

p. 11:

- line 314: "... techniques to observe vRNPs at different stages during replication."

p. 14:

- line 414: "... were carried out using Prism 8."

p. 17:

- line 454: "... FluPol and/or huANP32B. In a-e, data represent mean values ± s.e.m, based on n ≥ 3 independent ..."

p. 18: 

- line 465: "... and full-length or truncated ..."

- line 471: " In a and b, data represent mean values ± s.e.m, based on n ≥ 3 independent ..."

p. 21:

- line 485: "e, Effect of oligomerisation capacity and RNA-binding ability on the functions of NP in viral replication."

- line 489: "... significance. In a-e, data represent mean values ± s.e.m, based on n ≥ 3 independent ..."

p. 22: 

- line 504: "... modification, from a transcriptase to a replicase that initiates cRNA synthesis in a primer-independent manner (2)."

p. 26:

- line 533: "... NP proteins were purified from ..."

- line 534: move " * indicates NP degradation product." to the legend part describing panel a (line 533); also increase the size of the asterisk in Fig. S3a.

Reviewer #2: The paper "Defining the minimal components of the influenza A virus replication machinery via an in vitro reconstitution system" by Zhu et al. is well written and of high quality. I fully agree with the authors that this study is the first to establish a fascinating in vitro method to study the different vRNP activities.

There are some open questions that can be addressed by the authors:

1) As shown in Figure 3a and b, there is only a two-to-three-fold accumulation of vRNA in the presence of FluPol WT, NP and huANP32 but a >20 fold increase in cRNA level. Do the authors speculate that only very little functional cRNPs are generated? This should be discussed. 

2) Figure 3e: The monomeric NP mutant can still support cRNA/cRNP synthesis and in particular lead to an increase in vRNA levels. Though the authors quote an earlier paper regarding cRNA stabilization, however, full length vRNA accumulation seems to require NP-oligomerization (see also https://doi.org/10.1038/ncomms2589: e.g. Fig. 2a,). Is there an explanation for this discrepancy? 

3) Has the monomeric NP mutant (R416A) from A/NT/60/1968 been shown to behave as a monomer? If not, could the authors provide biochemical data on this?

4) Figure 3f: The authors state that new (and functional) vRNPs are formed in the presence of FluPol WT, NP and huANP32, but there is no direct evidence for this. Can the authors rule out that NP-free vRNA accumulates? If not, this possibility should be discussed.

5) Lane 406: Segment 5 encodes NP and not NA.

6) To show the robustness of this in vitro assay, it would be good to know the vRNP activity of a segment other than segment 5.

---

## [Editor Report · Decision Letter 2]

9 Oct 2023

Dear Dr Fodor,

Thank you for the submission of your revised Research Article "Defining the minimal components of the influenza A virus replication machinery via an in vitro reconstitution system" for publication in PLOS Biology. On behalf of my colleagues and the Academic Editor, Andrew Mehle, I am pleased to say that we can in principle accept your manuscript for publication, provided you address any remaining formatting and reporting issues. These will be detailed in an email you should receive within 2-3 business days from our colleagues in the journal operations team; no action is required from you until then. Please note that we will not be able to formally accept your manuscript and schedule it for publication until you have completed any requested changes.

PRESS

Sincerely, 

Paula

---

Senior Editor

PLOS Biology
